# Kallmann Syndrome: Functional Analysis of a *CHD7* Missense Variant Shows Aberrant RNA Splicing

**DOI:** 10.3390/ijms252212061

**Published:** 2024-11-10

**Authors:** Josianne Nunes Carriço, Catarina Inês Gonçalves, José Maria Aragüés, Manuel Carlos Lemos

**Affiliations:** 1CICS-UBI, Health Sciences Research Centre, University of Beira Interior, 6200-506 Covilhã, Portugal; 2Serviço de Endocrinologia, Diabetes e Metabolismo, Hospital de Santa Maria, Centro Hospitalar Universitário Lisboa Norte, 1649-028 Lisboa, Portugal

**Keywords:** Kallmann syndrome, congenital hypogonadotropic hypogonadism, fertility, CHD7, genetics, mutation, variant of uncertain significance, VUS, splicing

## Abstract

Kallmann syndrome is a rare disorder characterized by hypogonadotropic hypogonadism and an impaired sense of smell (anosmia or hyposmia) caused by congenital defects in the development of the gonadotropin-releasing hormone (GnRH) and olfactory neurons. Mutations in several genes have been associated with Kallmann syndrome. However, genetic testing of this disorder often reveals variants of uncertain significance (VUS) that remain uninterpreted without experimental validation. The aim of this study was to analyze the functional consequences of a heterozygous missense VUS in the *CHD7* gene (c.4354G>T, p.Val1452Leu), in a patient with Kallmann syndrome with reversal of hypogonadism. The variant, located in the first nucleotide of exon 19, was analyzed using minigene assays to determine its effect on ribonucleic acid (RNA) splicing. These showed that the variant generates two different transcripts: a full-length transcript with the missense change (p.Val1452Leu), and an abnormally spliced transcript lacking exon 19. The latter results in an in-frame deletion (p.Val1452_Lys1511del) that disrupts the helicase C-terminal domain of the CHD7 protein. The variant was reclassified as likely pathogenic. These findings demonstrate that missense variants can exert more extensive effects beyond simple amino acid substitutions and underscore the critical role of functional analyses in VUS reclassification and genetic diagnosis.

## 1. Introduction

Kallmann syndrome is a rare disorder characterized by hypogonadotropic hypogonadism and an impaired sense of smell (anosmia or hyposmia) due to congenital defects in the development of the gonadotropin-releasing hormone (GnRH) and olfactory neurons [1,2]. Affected individuals exhibit low or inappropriately normal levels of gonadotropins (luteinizing hormone (LH) and follicle-stimulating hormone (FSH)), and low levels of sex steroids, resulting in delayed or absent puberty and infertility [1,2]. The prevalence of Kallmann syndrome in different populations has been estimated to range from 1 in 10,000 to 1 in 86,000, with a male-to-female ratio of approximately four to one [2].

The genetic basis of Kallmann syndrome is highly heterogeneous, with incomplete penetrance, variable expressivity, and oligogenic inheritance contributing to its complexity [3,4,5]. Genetic testing frequently uncovers genetic variants of uncertain significance (VUS), where there is insufficient evidence to link the variant to the disorder. Distinguishing true causative variants from VUS that may be mistakenly classified as pathogenic is essential for reliable genetic diagnosis [6].

The chromodomain helicase DNA-binding protein 7 (*CHD7*) gene encodes a protein involved in chromatin remodeling, gene regulation, and embryonic development, particularly in the central nervous system, craniofacial structures, heart, and sensory organs [7]. While *CHD7* mutations were first linked to CHARGE (coloboma, heart defects, atresia choanae, retardation of growth and/or development, genital abnormalities, and ear abnormalities) syndrome [8], they have also been identified in patients with Kallmann syndrome and normosmic congenital hypogonadotropic hypogonadism [9]. However, *CHD7* mutations in Kallmann syndrome are generally milder than those associated with CHARGE syndrome [10].

Previously, we reported a Kallmann syndrome patient harboring a *CHD7* missense variant classified as a VUS due to insufficient evidence of pathogenicity [11]. The aim of this study was to elucidate the functional consequences of this variant and its role in the disorder.

## 2. Results

The patient presented a heterozygous missense variant (NM_017780.4: c.4354G>T, p.Val1452Leu) located in the first nucleotide of exon 19 (Figure 1A). Minigene experiments showed that the wild-type minigene produced a ribonucleic acid (RNA) transcript of the expected size (769 base-pair (bp)), including the properly spliced *CHD7* exon 19 (Figure 1B). In contrast, the mutant minigene generated two different transcripts: one with the expected 769 bp, containing the missense change (p.Val1452Leu), and another with 589 bp, missing exon 19 (Figure 1B). Densitometric analysis indicated that 59% of the mutant transcripts retained the missense change, while 41% exhibited exon skipping (Figure 1C). As exon 19 has 180 bp, its exclusion in the RNA transcript is predicted to lead to an in-frame deletion of 60 amino acids (p.Val1452_Lys1511del), affecting the CHD7 protein’s helicase C-terminal domain (Figure 1D). Based on these findings, the variant was reclassified as likely pathogenic according to the American College of Medical Genetics and Genomics (ACMG) criteria: PS3 (in vitro functional studies supportive of a damaging effect on the RNA), PM2 (absent from controls in population databases), and PM4 (protein length change due to in-frame deletion) [12].

## 3. Discussion

We identified a patient with Kallmann syndrome with a heterozygous missense variant in the *CHD7* gene (NM_017780.4: c.4354G>T, p.Val1452Leu) that was originally classified as a VUS [11]. Given the variant’s location at the first nucleotide of exon 19, we hypothesized and confirmed that it interferes with normal RNA splicing, leading to exon 19 skipping. This results in an in-frame deletion (p.Val1452_Lys1511del) affecting the CHD7 protein’s helicase C-terminal domain.

Splice-site variants are well-known contributors to human genetic disorders [13,14]. In particular, variants affecting the GT and AG dinucleotides at the 5′ and 3′ ends of introns, respectively, impair the recognition of splice-sites during the processing of RNA, resulting in aberrant RNA transcripts that can undergo nonsense-mediated decay or translate into abnormal proteins [13,14]. While the impact of variants affecting the first nucleotide of an exon is less studied, they can disrupt splicing depending on the length of the adjacent intronic polypyrimidine tract [15]. The presence of 15 or more consecutive pyrimidines ensures a normal splicing, despite the presence of the exonic variant, whereas the presence of 10 or less consecutive pyrimidines predisposes to aberrant splicing [15]. The polypyrimidine tract preceding *CHD7* exon 19 has only seven consecutive pyrimidines (TTTCTCT) and this may explain why the variant causes aberrant splicing.

Our study demonstrated that the missense variant had a disruptive effect on splicing. However, this effect was only partial, resulting in a mixture of full-length and exon-skipped transcripts, in a proportion of 59% and 41%, respectively. This indicates that the variant reduces the efficiency of exon 19 splicing without completely abolishing it. The full-length transcript is predicted to produce a CHD7 protein with a missense change at residue 1452 (p.Val1452Leu). However, this amino acid change is unlikely to be pathogenic by itself, as valine and leucine are similar non-polar branched-chain amino acids, and computational prediction tools [16,17,18] indicate a neutral effect of the amino acid change on the CHD7 protein. Furthermore, there have been no published reports of other amino acid changes at this location. This suggests that the pathogenic effect of the variant is not due to the amino acid change, but rather to the disruptive effect on RNA splicing. This disruptive effect of the variant may be partially mitigated by the production of a functional missense protein, leading to a milder overall impact compared to typical splice-site mutations. This may explain why our patient had only hypogonadism without the more complex phenotype of CHARGE syndrome that is usually associated with more damaging *CHD7* mutations [10]. Furthermore, our patient had reversal of the hypogonadism at the age of 33, suggesting a less severe impact of the mutation on the reproductive phenotype. Reversal of hypogonadotropic hypogonadism after treatment discontinuation has been reported in approximately 10% of patients and is often associated with a milder reproductive phenotype [19,20]. When reversal occurs, the body begins to produce sufficient levels of gonadotropins and sex hormones without the need for hormone replacement therapy. The exact cause of this reversal is not entirely understood, but it is thought that genetic factors, environmental influences, and even specific aspects of treatment may contribute to the reactivation of the hypothalamus–pituitary–gonadal axis in some patients [19,20]. Altogether, our functional studies suggest a possible explanation for the association between the *CHD7* variant and the patient phenotype.

The reclassification of this *CHD7* variant from VUS to likely pathogenic underscores the importance of functional studies in genetic diagnosis. VUS are common findings in genetic testing but often remain uninterpreted without experimental validation [6]. Our findings highlight the value of such analyses in refining the clinical significance of genetic variants.

A limitation of our study is the reliance on in vitro assays, which may not fully replicate the natural expression and splicing conditions in human tissues. Furthermore, functional assessments at the protein level were not conducted, leaving some uncertainty about the effects of both the missense change and the in-frame deletion on CHD7 protein function.

In conclusion, our study of a patient with Kallmann syndrome uncovered an unusual missense mutation in the *CHD7* gene that partially disrupts normal RNA splicing. The functional analysis of VUS, such as the one described here, is crucial for advancing our understanding of their role in human disease and improving genetic diagnosis.

## 4. Materials and Methods

### 4.1. Clinical Case

A 21-year-old man presented with delayed puberty and anosmia. His medical history was unremarkable, with no family history of delayed or absent puberty. Physical examination revealed scant body hair and genital hypoplasia without other CHARGE syndrome features, namely, coloboma, heart defects, atresia of choanae, retardation of growth, or ear abnormalities. Blood tests showed low total testosterone (0.7 ng/mL; normal: 2.6–16.0), FSH (1.7 mIU/mL; normal: 1.5–12.0), and LH (0.9 mIU/mL; normal: 1.0–8.5). A GnRH test was not performed. Other pituitary hormone levels and brain magnetic resonance imaging (MRI) were normal. The clinical presentation of anosmia and hypogonadotropic hypogonadism (low levels of serum testosterone, FSH, and LH) led to a diagnosis of Kallmann syndrome [1,2]. Testosterone replacement therapy was initiated, resulting in the development of secondary sexual characteristics. The patient was then maintained on monthly intramuscular injections with regular monitoring of serum testosterone. After 6 to 7 years, the frequency of injections was reduced as the patient was able to sustain normal testosterone levels for longer intervals. By age 33, testosterone therapy was discontinued entirely, and the patient maintained normal serum testosterone without treatment, indicating a reversal of the hypogonadism. The genetic testing of this patient has already been reported [11], revealing a heterozygous missense variant in the *CHD7* gene (NM_017780.4: c.4354G>T, p.Val1452Leu). The variant was also present in his apparently unaffected father and absent in his mother. This variant affected an evolutionary conserved nucleotide [21] and was absent from the gnomAD population database [22]. However, computational tools such as SIFT (Sorting Tolerant From Intolerant) [16], PolyPhen-2 [17], and AlphaMissense [18] predicted that the amino acid change had a neutral effect on the protein. According to ACMG criteria [12], there was insufficient evidence for a pathogenic role of the variant, and therefore the variant was classified as a VUS.

### 4.2. Functional Studies of the Missense Variant

Given the variant’s location at the first nucleotide of exon 19 (Figure 1A), we hypothesized that it might disrupt splicing at the adjacent splice-site. Since patient RNA was unavailable, we constructed an artificial minigene using previously described methods [23]. In short, wild-type (normal) and mutant *CHD7* exon 19 sequences, including the 5′ (231 bp) and 3′ (439 bp) intronic flanking regions, were obtained from patient genomic deoxyribonucleic acid (DNA) by polymerase chain reaction (PCR) and cloned into the NdeI and BglII restriction sites of the pcAT7-Glo1 plasmid (a kind gift from Dr. Kristen W. Lynch, Perelman School of Medicine, University of Pennsylvania, USA) (Figure 1A). The minigene plasmid DNA was used to transfect COS-7 cells and the total RNA was then extracted and used to synthesize complementary DNA (cDNA). This was then used as a template for PCR amplification with primers Act and ActT7R [23] (Figure 1A). PCR of the GloE3 exon was used as an internal control. The PCR amplified products were then run on an electrophoresis gel to determine the size and density of products. The products were then excised from the gel and sequenced using a semi-automated DNA sequencer (STAB VIDA, Caparica, Portugal; and ABI 3730XL, Applied Biosystems; Thermo Fisher Scientific, Waltham, MA, USA). To visualize the affected domains of the CHD7 protein, a three-dimensional model was generated using the SWISS-MODEL web-based integrated service and a DNA helicase template (id A0A670ZHI7) [24].

## Figures and Tables

**Figure 1 ijms-25-12061-f001:**
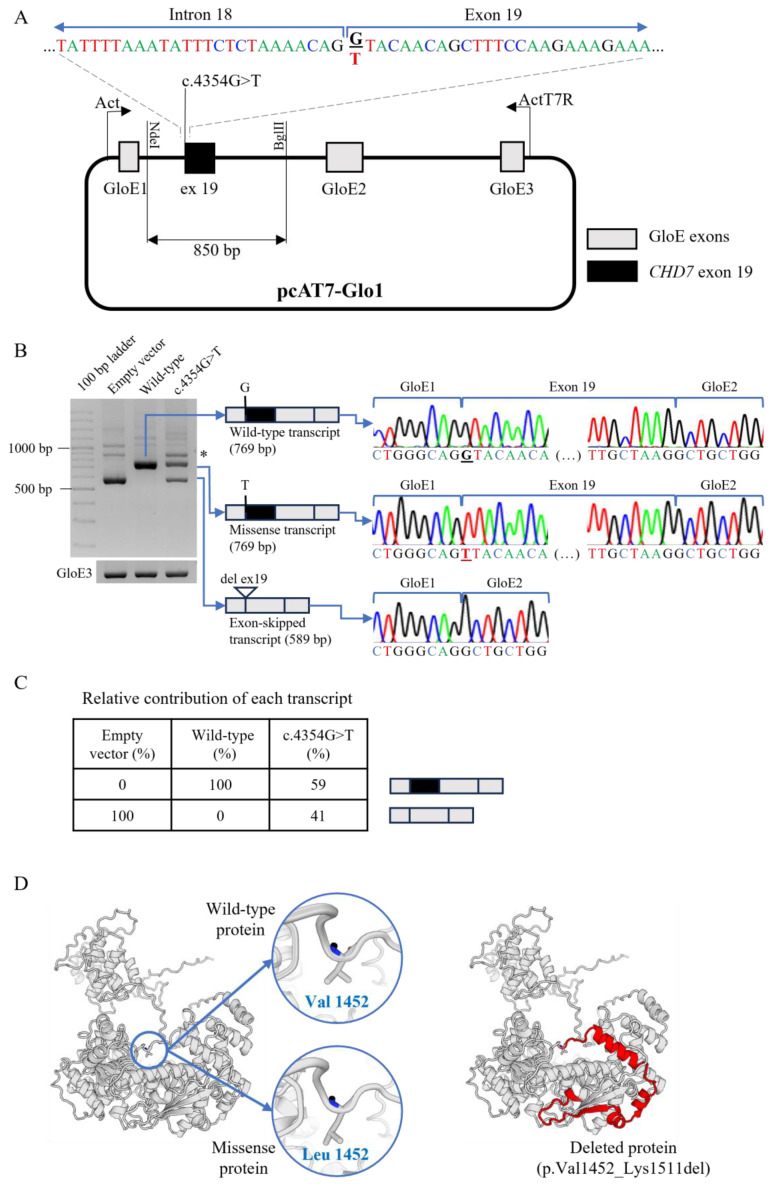
Functional analysis of the *CHD7* missense variant. (**A**) The patient harbored a heterozygous missense variant in the *CHD7* gene (c.4354G>T), located at the first nucleotide of exon 19 (highlighted in bold). An 850-base-pair (bp) genomic fragment from the patient, encompassing exon 19 (with and without the variant) and flanking intronic sequences, was cloned into the pcAT7-Glo1 minigene vector (containing a modified version of the human β-globin gene) at the NdeI and BglII restriction sites, between globin exons 1 (GloE1) and 2 (GloE2). The constructed minigene plasmid was transfected into COS-7 cells, and the resulting ribonucleic acid (RNA) was analyzed using reverse transcriptase polymerase chain reaction (RT-PCR) with primers Act and ActT7R. (**B**) Electrophoresis and sequencing of the amplified complementary deoxyribonucleic acid (cDNA) fragments showed that the wild-type allele generated a single, normally spliced transcript (769 bp). In contrast, the mutant allele produced two distinct transcripts: one corresponding to the normally spliced transcript containing the missense variant (769 bp), and another with an aberrantly spliced transcript with a deletion (del) of exon 19 (589 bp). PCR amplification of the GloE3 exon served as an internal control. The asterisk indicates heteroduplex fragments. (**C**) Densitometric analysis of the amplified cDNA demonstrated that the wild-type allele exclusively generated normally spliced transcripts (100%), while the mutant allele expressed both the missense and exon-skipped transcripts in proportions of 59% and 41%, respectively. (**D**) SWISS-MODEL-generated three-dimensional representation of the CHD7 protein showing the wild-type (normal) protein containing a valine at position 1452, the missense protein containing a leucine at this position, and the deleted protein which lacks a sequence of 60 amino acids (in red).

## Data Availability

The data from this study are available from the corresponding author on reasonable request.

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
