# Peer review of "Kallmann Syndrome: Functional Analysis of a CHD7 Missense Variant Shows Aberrant RNA Splicing"

_ijms, 2024, doi:10.3390/ijms252212061_

Round 1
Reviewer 1 Report (New Reviewer)
Comments and Suggestions for Authors
The submitted manuscript is a (very) brief report on the functional analysis of a CHD7 Missense Variant associated with Kallmann Syndrome.
The Authors have previously reported a Kallmann syndrome patient with a CHD7 missense mutation classified as a variant of uncertain significance due to inadequate evidence of harm, as described in their work [11]. The objective of this study was to clarify the functional implications of this variation and its involvement in the disease.
Personally, I’m against the “short communication” type of article, as I have a feeling that they could have been elaborated more. However, I understand that in this case it is very unlikely to increase the population size. As the work is generally well written and presented, it can be accepted after several revisions.
1. Line 29, Please add some information on the epidemiology (numbers)
2. Line 67, a reference is needed here
3. Figure 1D, how this figure was obtained?
4. Line147, “without other CHARGE syndrome features” – such as?
5. Line 148, FSH is OK.
6. Lines 167-169, this should be moved to Lines 201-202.
7. Lines 87-88, have the other missense protein, with Val1542 replaced by other aa, been reported in the literature?
8. Lines 114-117, while the mention tools perform the predictions based on previously published models, it would be worth to optimize such structure using molecular modeling methods to confirm this hypothesis.
Author Response
The submitted manuscript is a (very) brief report on the functional analysis of a CHD7 Missense Variant associated with Kallmann Syndrome.
The Authors have previously reported a Kallmann syndrome patient with a CHD7 missense mutation classified as a variant of uncertain significance due to inadequate evidence of harm, as described in their work [11]. The objective of this study was to clarify the functional implications of this variation and its involvement in the disease.
Personally, I’m against the “short communication” type of article, as I have a feeling that they could have been elaborated more. However, I understand that in this case it is very unlikely to increase the population size. As the work is generally well written and presented, it can be accepted after several revisions.
>>>Authors’ response: We are grateful to the reviewer for his/her comments as they have helped to improve the manuscript. Please find below a point-by-point response to the comments. We have made changes to the text of the manuscript and have uploaded a copy with the marked changes along with a clean copy.
- Line 29, Please add some information on the epidemiology (numbers)
>>>Authors’ response: We have added values for prevalence and male-to-female ratio.
- Line 67, a reference is needed here
>>>Authors’ response: We have added the reference for the ACMG classification.
- Figure 1D, how this figure was obtained?
>>>Authors’ response: We have added “SWISS-MODEL-generated three-dimensional representation (…)” in the figure legend. Further details are presented in the methods section.
- Line147, “without other CHARGE syndrome features” – such as?
>>>Authors’ response: We have specified the CHARGE features which were not present in the patient.
- Line 148, FSH is OK.
>>>Authors’ response: Yes, this is correct. The FSH level is within the normal population range, but inappropriate for the reduced testosterone levels (due to the inability of the hypothalamus/pituitary to produce a feedback response). As stated in the first paragraph of the introduction, affected individuals can exhibit low or inappropriately normal levels of gonadotropins.
- Lines 167-169, this should be moved to Lines 201-202.
>>>Authors’ response: Thank you for your suggestion. We have deleted the ethics and informed consent statements from the methods, as these are already included at the end of the manuscript.
- Lines 87-88, have the other missense protein, with Val1542 replaced by other aa, been reported in the literature?
>>>Authors’ response: We have found no other reports of aa changes at this position. We have added this important information in the third paragraph of the discussion.
- Lines 114-117, while the mention tools perform the predictions based on previously published models, it would be worth to optimize such structure using molecular modeling methods to confirm this hypothesis.
>>>Authors’ response: We applied the widely used PolyPhen and SIFT computational tools to predict the effect of the amino acid substitution. These predicted that the amino acid substitution had no functional effect on the protein. Unfortunately, we are currently unable to further investigate this through molecular modelling, as there is limited availability of three-dimensional structural data for CHD7. We apologize for not being able to carry this out. However, to validate the predictions by PolyPhen and SIFT, we have now included an additional analysis using the AlphaMissense tool that confirmed a benign effect of the amino acid substitution (we added the relevant reference in the methods and discussion). We hope that this is an acceptable compromise.
Reviewer 2 Report (New Reviewer)
Comments and Suggestions for Authors
Lines 66–67: Please provide a more detailed explanation of how this variant meets each criterion, and cite the 2015 guideline.
Author Response
Lines 66–67: Please provide a more detailed explanation of how this variant meets each criterion, and cite the 2015 guideline.
>>>Authors’ response: Thank you for this useful suggestion. We have updated and explained the criteria and provided the supporting reference.
Reviewer 3 Report (New Reviewer)
Comments and Suggestions for Authors
The brief report involves the description of a mutation of the CDCH7 gene. The patient is heterozygote for the mutation. However, it is unclear from the description why the clinical manifestation was hypogonadism, and after years of treatment with testosterone, the patient can maintain its levels. The authors should provide a valid explanation of the phenomenon based on their clinical and biological information on hands.
The missense mutation is interesting on one site. Still, unless a functional analysis is performed, it would be hard to convince the reader that changing leucine for valine will modify the protein's tertiary structure. Since the authors did hard work of subcloning, I would recommend using this method DOI: 10.1038/s41431-019-0465-7
I also recommend that the authors add the limitations of the study.
Author Response
The brief report involves the description of a mutation of the CHD7 gene.
>>>Authors’ response: We are grateful to the reviewer for his/her comments and for the opportunity to improve our manuscript. We have made changes to the text of the manuscript and have uploaded a copy with the marked changes along with a clean copy.
The patient is heterozygote for the mutation. However, it is unclear from the description why the clinical manifestation was hypogonadism, and after years of treatment with testosterone, the patient can maintain its levels. The authors should provide a valid explanation of the phenomenon based on their clinical and biological information on hands.
>>>Authors’ response: Thank you for raising this important point. Reversal of hypogonadism in patients can be surprising but is not uncommon (reported in approximately 10% of cases). The exact cause of this reversal is not entirely understood, but it is thought that genetic factors, environmental influences, and even specific aspects of treatment may contribute to the reactivation of the hypothalamus-pituitary-gonadal axis in some patients. We have completed the discussion with this explanation.
The missense mutation is interesting on one site. Still, unless a functional analysis is performed, it would be hard to convince the reader that changing leucine for valine will modify the protein's tertiary structure. Since the authors did hard work of subcloning, I would recommend using this method DOI: 10.1038/s41431-019-0465-7
>>>Authors’ response: We agree that this would be an interesting addition to the study but unfortunately, we are currently unable to undertake this. Functional studies of the CHD7 protein are technically challenging and very rarely carried out. In fact, since the identification of the first CHD7 mutations in 2004, there have been hundreds of published mutations. Apart from a single report (cited by the reviewer) that has not been replicated by anyone else in the field, there are no other studies of CHD7 variants at the protein level. Even splice-site mutations are rarely studied at the RNA level. Our study goes further than most studies in the field. Even if we were to spend the next months or years developing a protein functional assay, the results would not significantly influence our conclusions, because the effect of our variant at the RNA level is sufficient for the purpose of genetic diagnosis, according to current ACMG criteria (Richards et al. PMID: 25741868). We do not claim that changing leucine for valine will modify the protein's tertiary structure. On the contrary, we provide evidence that this amino acid change is neutral and that the variant exerts its pathogenic effect through an entirely different mechanism (exon skipping with the loss of 60 amino acids). Thus, an important message of the manuscript is that missense variants can sometimes exert more extensive effects beyond simple amino acid substitutions, and we believe that this constitutes valuable information for the readers of the article. We apologize to the reviewer for not undertaking this and we hope that he/she will accept our explanation.
I also recommend that the authors add the limitations of the study.
>>>Authors’ response: We have revised the limitations of the study (paragraph before conclusions).
Round 2
Reviewer 1 Report (New Reviewer)
Comments and Suggestions for Authors
I'm satisfied with the corrections and improvements done by the authors.
Reviewer 3 Report (New Reviewer)
Comments and Suggestions for Authors
The authors modified the text. I agree, in part, with the explanation of the deletion's importance within the protein. However, my point is crucial functionality. I understand the group's limitations in pursuing the experiments, but the patient's hypogonadism cannot be explained by the mutation, in my opinion.
This manuscript is a resubmission of an earlier submission. The following is a list of the peer review reports and author responses from that submission.
Round 1
Reviewer 1 Report
Comments and Suggestions for Authors
This manuscript was expected to present the functional consequences of the VUS missense mutation in the CHD7 gene, but in reality, it has a major limitation in that the functional parts are presented only as fragmentary facts. The topic of this manuscript is interesting. However, there are clearly controversial and uncertain parts. This manuscript definitely needs clarification and interpretation of additional parts. I do not think this manuscript is ready for publication at this stage. This manuscript needs to be supplemented with additional studies to secure at least a minimum of clarity.
Minor points:
Line 47: Additional explanation of the meaning of VUS is required. This will help readers understand.
Figure 1: Full name of ‘GloE’, ‘pcAT7Glo1’ is required.
Major/Critical points (Sufficient explanation and experimental reinforcement are required):
The generation of ‘p.Val1452Leu’ is understood to be an amino acid substitution due to missense mutation (c.4354G>T). However, what is the mechanism by which ‘p.Val1452_Lys1511del’ is generated? In other words, a mechanistic explanation is required for the generation of two different transcripts by a single genetic mutation (c.4354G>T).
Line of 100-102: A clear mechanistic interpretation of how different ratios of transcripts are generated by a single missense mutation (c.4354G>T) is required.
Line of 114-117: I think this is a limitation of the manuscript and not a problem that can be solved (in current manuscript state). This manuscript must confirm the functional aspects at the protein level through additional experiments and present the results together. No meaningful genetic significance (c.4354G>T) interpretation can be made without connecting the protein functional aspects.
In addition, this study does not suggest any associations with the pathological and clinical symptoms of CHD7 gene c.4354G>T. This strongly suggests that the current manuscript cannot provide any certainty.
Comments on the Quality of English LanguageThe English needs to be polished a bit more.
Reviewer 2 Report
Comments and Suggestions for Authors
Thank you for the opportunity to review the manuscript “Kallmann Syndrome: Functional Analysis of a CHD7 Missense Variant Shows Aberrant RNA Splicing” by Josianne Nunes Carriço et al. This is a case study of a novel genetic cause of Kallmann Syndrome. However, the clinical description of the patient is not explicit. Was the GnRH test performed at the diagnosis of hypogonadotropic hypogonadism? What does hypogonadism reversal mean?
Comments on the Quality of English LanguageMinor editing of English language required.
Round 2
Reviewer 1 Report
Comments and Suggestions for Authors
The authors' responses to the reviewer comments are very insincere and it is difficult to determine which part was specifically answered.
And above all, I have already decided that this manuscript cannot be published at this stage.
The reason is that the results in terms of the functional aspect of the gene have not been confirmed at all.
The authors have not provided a clear answer to the reviewer comments on this part.
I strongly oppose publishing this manuscript at this stage.
I also refuse to review the manuscript without further experimental data and clear interpretations.
Comments on the Quality of English LanguageThe English text is still rough and unreadable.
Author Response
Comments and Suggestions for Authors
The authors' responses to the reviewer comments are very insincere and it is difficult to determine which part was specifically answered.
And above all, I have already decided that this manuscript cannot be published at this stage.
The reason is that the results in terms of the functional aspect of the gene have not been confirmed at all.
The authors have not provided a clear answer to the reviewer comments on this part.
I strongly oppose publishing this manuscript at this stage.
I also refuse to review the manuscript without further experimental data and clear interpretations.
Comments on the Quality of English Language: The English text is still rough and unreadable.
>>>>Author response: The reviewer requests additional experiments at the protein level which unfortunately are not possible to perform. This is because of the lack of reliable functional assays for this particular protein (CHD7). In fact, since the identification of the first CHD7 mutations in 2004, there have been hundreds of published mutations. Apart from a single report (Brajadenta et al. PMID: 31289371) that has not been replicated by anyone else in the field, there are no studies of CHD7 variants at the protein level. Even splice-site mutations are rarely studied at the RNA level. Our study goes further than most studies in the field. Even if we were to spend the next months or years developing a protein functional assay, the results would not significantly influence our conclusions, because the effect of our variant at the RNA level is sufficient for the purpose of genetic diagnosis, according to current ACMG criteria (Richards et al. PMID: 25741868). An important message of the manuscript is that missense variants can sometimes exert more extensive effects beyond simple amino acid substitutions, and we believe that this constitutes valuable information for the readers of the article.
As an alternative to the protein experiments, which cannot currently be performed, we have added the results of CHD7 protein modelling using SWISS-MODEL that may allow the readers to better understand the predicted effects of the variant on the protein structure (Figure 1 D). We do hope this is a suitable compromise. We have also clarified some parts of the text, specially those related to the interpretation of the data. All changes, including those made in the previous round of revisions, are marked in the document.
Reviewer 2 Report
Comments and Suggestions for Authors
Although the report is novel and interesting, the clinical case presented by the Authors lacks some important diagnostic issues, e.g. GnRH test. Therefore the described hypogonadism and then hypogonadism reversal seems not to be certain. The genetic tests must be reviewed by a geneticist.
Comments on the Quality of English LanguageMinor editing of English language required.
Author Response
Comments and Suggestions for Authors
Although the report is novel and interesting, the clinical case presented by the Authors lacks some important diagnostic issues, e.g. GnRH test. Therefore the described hypogonadism and then hypogonadism reversal seems not to be certain. The genetic tests must be reviewed by a geneticist.
Comments on the Quality of English Language: Minor editing of English language required.
>>>>Author response: Thank you again for your time in reviewing this manuscript. We understand your reservations and we agree that the diagnosis of congenital hypogonadotropic hypogonadism (CHH) is often challenging and frequently confused with other situations such as constitutional delay of growth and puberty (CDGP). Unfortunately, to date, no gold-standard diagnostic test exists to fully differentiate CHH from CDGP. Even the GnRH test has been considered to have questionable value for the diagnosis of CHH (Harrington et al. PMID: 22723321; Boehm et al. PMID: 26194704; Young et al. PMID: 30698671). Our patient was diagnosed using standard criteria which are generally accepted in the field (Boehm et al. PMID: 26194704; Young et al. PMID: 30698671). The association of anosmia with CHH leaves little doubt about the diagnosis of Kallmann syndrome. If there were still any doubts about the diagnosis, this was finally resolved with the genetic results. In the clinical description of the patient, we have now added references that support the diagnosis of the patient. We hope that the reviewer agrees with this. We would also like to confirm that all genetic results from our lab are reviewed by experts in genetics.